# A Real-Life Study on the Use of Tildrakizumab in Psoriatic Patients

**DOI:** 10.3390/ph16040526

**Published:** 2023-03-31

**Authors:** Elena Campione, Sara Lambiase, Ruslana Gaeta Shumak, Marco Galluzzo, Caterina Lanna, Gaetana Costanza, Cristiana Borselli, Fabio Artosi, Terenzio Cosio, Lorenzo Tofani, Annunziata Dattola, Francesca Di Daniele, Luca Bianchi

**Affiliations:** 1Dermatology Unit, Department of Systems Medicine, University of Rome Tor Vergata, 00133 Rome, Italy; 2Department of Experimental Medicine, University of Rome Tor Vergata, 00133 Rome, Italy

**Keywords:** tildrakizumab, IL-17, psoriasis, comorbidities, PASI, NAPSI, PPPGA, DLQI

## Abstract

Tildrakizumab is a humanized IgG1κ monoclonal antibody that selectively targets the p19 subunit of interleukin IL-23, thereby inhibiting the IL-23/IL-17 axis, which is primarily implicated in the immunopathogenesis of psoriasis. Tildrakizumab is approved for the treatment of moderate-to-severe plaque-type psoriasis in adults based on the evidence of two randomized and controlled phase-III clinical trials (reSURFACE 1 and reSURFACE 2). Here, we report our real-life experience treating 53 psoriatic patients (19 female and 34 male) who were administered tildrakizumab every 12 weeks and received follow-ups over 52 weeks. Descriptive and inferential statistical analyses were performed, in particular the Psoriasis Area and Severity Index (PASI), Dermatology Life Quality Index (DLQI) and, if applicable, the Nail Psoriasis Severity Index (NAPSI) and Palmoplantar Psoriasis Physician Global Assessment (PPPGA). These were assessed at baseline and after different timepoints (weeks) during the follow-up period. We described and evaluated demographical and epidemiological characteristics in our cohort group, focusing on comorbidities. In this group, 35.9% of patients were female and 64.1% were male, with 47.1% being smokers and with a mean age of 51.2 years. A total of 37.7% of these patients was affected by scalp psoriasis; regarding comorbidities, hypertension was the most frequent (32.5%), followed by psoriatic arthritis (PsA) (18.60%) and diabetes (13.9%). At week 52, 93%, 90.2% and 77% of patients achieved a PASI reduction ≥75% (PASI 75), PASI 90 and PASI 100, respectively. In addition, NAPSI, PPPGA and DLQI scores were significantly reduced by week 52. In our cohort of complex psoriasis patients, disease remission began at the end of the fourth week of treatment and remained constant from week 16 to week 52.

## 1. Introduction

Psoriasis is a chronic, immune-mediated skin disease that affects approximately 125 million people worldwide [1]. Psoriasis most commonly manifests itself as a skin disease, consisting of well-demarcated, erythematous scaly plaques that can occur anywhere on the body, but particularly on the elbows, knees, scalp and lumbosacral area [2]. In the initial steps of psoriasis pathogenesis, a variety of cell types, including plasmacytoid dendritic cells, keratinocytes, natural killer T cells and macrophages, secrete cytokines that activate myeloid dendritic cells [2,3]. Psoriasis can also substantially impair the quality of life and the mental wellbeing of patients [3]. IL-23 supports the survival of Th17 cells, producing downstream cytokines including IL-17, which stimulates the production of pro-inflammatory cytokines and chemokines, resulting in systemic inflammation and joint injury. The increased expression of both IL-23 and IL-17 in psoriatic plaques compared to normal skin highlights the pivotal role of IL-23 in the pathogenesis of psoriasis and psoriatic arthritis. IL-23 is a heterodimer composed of two subunits: p19 and p40. The p40 subunit is shared by IL-12 and IL-23, whereas the p19 subunit is unique to IL-23 [4,5,6]. Plaque psoriasis is the most common variant and is characterized by sharply demarcated erythematous scaly patches or plaques that occur commonly on extensor surfaces, though it can also affect the intertriginous areas, palms, soles, scalp and nails [7,8]. The specific modifications of immune response as well as differences in immune cell types were studied with a focus on the regulation of immune genes in the palmoplantar keratinocyte pathway. Recently, Wiedemann et al. demonstrated that palmoplantar keratinocytes exhibit an increased expression of oxidative phosphorylation-related genes and a decreased expression of immune-related genes [9].

Although skin involvement is often the most prominent (and might be the only) manifestation of this disease, recognizing the condition as a chronic, multisystem inflammatory disorder, is imperative to optimize its management [5,6]. In fact, psoriatic patients have a higher prevalence of depression, obesity, cardiovascular disease (CVD) and metabolic syndrome (MetS) compared to the general population [10]. For patients with mild psoriasis, treatment options are represented by topical corticosteroids, vitamin D analogues, calcineurin inhibitors, keratolytics and targeted phototherapy. Moderate to severe psoriasis is treated with systemic agents and biologics. The four classes of biologics used to treat psoriasis are TNF inhibitors, IL-12/23 inhibitors, IL-17 inhibitors and IL-23 inhibitors [3,4,5,11]. Tildrakizumab is a high-affinity, humanized IgG1κ monoclonal antibody targeting the p19 subunit of IL-23, which is a key regulatory cytokine in psoriasis and stimulates the differentiation, proliferation and survival of T helper 17 cells. The efficacy and safety of tildrakizumab were evaluated in two double-blind, randomized controlled trials, reSURFACE 1 (NCT01722331) and reSURFACE 2 (NCT01729754), in comparison with a placebo and etanercept. High levels of efficacy were maintained for up to 3 years of psoriasis treatment with tildrakizumab [12,13]. There is a favorable long-term safety profile with tildrakizumab both in doses of 100 mg and 200 mg, with a low incidence of adverse events of special interest over 3 years [12,13]. Over 80% of patients entering extensions completed treatment through week 244 [14]. On the basis of the safety profile of the drug, which is also confirmed by European guidelines (EuroGuiDerm), its use is permitted to special population patients such as those with inflammatory bowel disease, cardiovascular disease, metabolic syndrome and advanced age [15,16]. Pooled data from two phase-III, double-blind, randomized, placebo-controlled studies (reSURFACE 1 (NCT01722331) and 2 (NCT01729754)) evaluated tildrakizumab’s efficacy, durability of response, and safety in patients with moderate to severe chronic psoriasis with and without MetS. The percentages of patients with ≥75% improvement on the Psorasis Area and Severity Index (PASI75 responders) at weeks 12 and 52 were comparable between patients with and without MetS for both tildrakizumab doses. Percentages of PASI 90 and PASI 100 responders through week 52 were similar regardless of MetS status for both tildrakizumab doses. Tildrakizumab efficacy through week 52 was maintained comparably in patients with and without MetS [14,15,16]. Percentages of patients with one or more serious adverse events (AEs) and those with one or more serious infections was similar in patients with and without MetS [10,15]. Here, we report our clinical real-life experience with 53 (19 women and 34 men) patients affected by moderate to severe psoriasis and several comorbidities, with follow-ups over 52 weeks, to evaluate the efficacy and safety of tildrakizumab. Different psoriatic scores were analyzed at baseline and at different timepoints (weeks = W). We even explored the improvement in difficult-to-treat areas, such as scalp, nails, palmoplantar and genital areas, to assess tildrakizumab’s efficacy in these sites.

## 2. Results

### 2.1. Demographic Characteristics of Enrolled Patients

The clinical and demographic characteristics of our 53 patients (mean age 51.2 years) recorded at baseline (W0) are shown in Table 1. In particular, 35.9% were female and 64.1% were male, with 47.1% being smokers. The mean age of disease onset was 34.6 years old. Patients were affected by moderate-to-severe plaque-type psoriasis with a mean PASI score of 13.6 (range 0–48) and a mean DLQI score of 10.2 (range 2–20) at baseline. Patients showed a body mass index (BMI) mean value of >27.2 (pre-obesity status), whereas 12 of 53 enrolled patients were overweight (BMI 26–29.99) and 16 were obese with a BMI > 30 (range 30–39). Furthermore, only eight patients (18.6%) had concomitant psoriatic arthritis (PsA).

Patients’ previous anti-psoriatic treatments are reported in Table 2; cyclosporine treatment was prevalent (27.3%). Some patients used more than one treatment.

We also divided the subjects with moderate-to-severe psoriasis into two groups: those who had undergone previous biological therapies (52.8%) and the naïve group (47.2%).

In addition, stratification was carried out for previous treatments: 9.61% of subjects started therapy with tildrakizumab as their first treatment, 15.1% had previously received only one treatment, 20.1% had received two other treatments, 20.7% had received three treatments and 33.9% had received four or more treatments. Thus, almost half of the patients enrolled underwent multiple treatments without achieving lasting effects over time.

The involvement of difficult-to-treat sites was also evaluated: 37.7% of patients were affected by scalp psoriasis, 15.1% had nail manifestations, 26.1% had palmoplantar psoriasis and a further 21.1% had genital involvement.

Patients’ comorbidities (number of patients = 43) were also recorded, with blood hypertension as the most frequent (32.5%), followed by psoriatic arthritis (PsA) (18.60%), diabetes (13.9%), infections (7%), neurological disorders (7%), hypercholesterolemia (4.65%), inflammatory bowel diseases (4.65%) and allergies (4.65%), whereas 7% were affected by other pathologies (Figure 1). Ten patients did not declare comorbidities.

### 2.2. Evaluation of PASI, PPPGA and DLQL Indexes

To define and investigate the efficacy of tildrakizumab on psoriatic plaques, we evaluated PASI at different timepoints (week = W). Statistical analysis showed a significant reduction of PASI (mean values) after 4 W of treatment, and the clinical response was maintained up to 52 W, as reported in Figure 2 (ANOVA test; *p* < 0.0001).

Significant clinical improvement was also confirmed by the proportion of patients achieving PASI75 (75% or greater improvement in PASI score), PASI 90 (90% or greater improvement in PASI score) and PASI 100 (100% improvement in PASI score) from baseline and after treatment, as reported in Figure 3. PASI75 was 77% at W16, 79% at W28 and 93% at W52. PASI90 was 61.5%, 67.8% and 90.2% at W16, W28 and W52, respectively. The percentages of patients achieving PASI100 were 48.7%, 57.1% and 77% at W16, W28 and W52, respectively.

The mean NAPSI score during the entire period of enrollment changed from 10.8 at T0 to 8 at W4 and to 0.8 by W52 (Figure 4 up; ANOVA test; *p* < 0.001). The mean PPPGA score also decreased from 4.4 at T0 to 4.1 at W4 and to 0 by W40 and W52, as reported in Figure 4 (down, ANOVA test; *p* < 0.0001).

Considering improvement in terms of quality of life as expressed by the DLQI, we also found a statistically significant reduction of symptoms after 16 weeks of treatment (Figure 5 ANOVA test; *p* < 0.001). Interestingly, patients with DLQI scores >10 also showed BMI scores >27, and 50% of them were affected by diabetes at baseline.

Figure 6 shows representative photographs of one patient, who showed a DLQI > 10 and a BMI > 27 at timepoint 0, before and after treatment, considering all indices discussed and to illustrate the efficacy of tildrakizumab after 28 weeks of treatment. Thus, the patient’s DLQI index was <5 after 28 weeks of the tildrakizumab therapeutic regimen.

### 2.3. Stratification of Patients Considering the Age of Onset of Psoriasis, PASI, and the Involvement of Difficult-to-Treat Sites

Based on the age of onset of psoriasis (34.6 years), we divided our patients into two groups and evaluated PASI at T0 as well as the involvement of difficult-to-treat sites and the presence of comorbidities. As shown in Figure 7, the first group consisted of patients with a psoriasis age of onset < 35 years, most of whom were men (60% M), of whom 80% had manifestations in the scalp and genital area and only 20% had comorbidities; their PASI at T0 was 13.3.

The second group had an age of onset >35 years, were also predominantly male (63% M) and difficult-to-treat sites were less involved (30% in the scalp and 20% in other sites); of these subjects, 65% (rho = 0.67) were affected by comorbidities.

### 2.4. Stratification by Comorbidities and Drug Safety Profile

We stratified the patient population into two groups based on the presence or absence of comorbidities by evaluating the percentage decrease in PASI at different weeks of treatment. As shown in Figure 8, the fit between the two subgroups analyzed is quite similar (ANOVA test; *p* < 0.01), except during week 4, when patients with comorbidities showed a lower decrease in mean PASI compared to patients without comorbidities (*t*-test; *p* < 0.01).

We also evaluated NAPSI and PPPGA, but no differences were observed between the two groups (the score decreases were similar for two groups). DLQI scores were also compared, and no differences were found.

Regarding the safety profile of the drug, during the 52 weeks of treatment, none of the patients presented adverse events, and only four of them discontinued the treatment due to primary inefficacy. In the present study, 34/53 patients reached treatment W52. Therefore, the presence of comorbidities does not affect the safety of this treatment.

## 3. Discussion

The value of a real-life clinical study of psoriasis patients also affected by other dysmetabolic conditions is that it could add new insights to the evolving scenario of this chronic inflammatory skin disease. Our study focused on the effects of the use of tildrakizumab, a humanized IgG1κ monoclonal antibody targeting the p19 subunit of IL-23, in a cohort of 53 patients affected by moderate-to-severe psoriasis. PASI, NAPSI, PPPGA and DLQI indexes were evaluated at different timepoints of enrollment along with the hematochemical parameters in the group of complex patients who also suffered from comorbidities. The crucial role of IL-23 in the cytokines cascade, including its responsibility for the clinical manifestations of psoriasis and the importance of its inhibition as a pathogenetic target in the resolution of the disease itself, emerges from data in the literature and from our real-life clinical experience [3,12,13,14,15,16]. Differently from IL-12/23 inhibitors, tildrakizumab does not interfere with the IL-12 pathway, thereby allowing one to preserve immunological surveillance. Several studies clearly illustrated the importance of endogenous IL-12 and IFN-γ in preventing cancer initiation, growth and metastasis. In contrast, IL-23 plays an important role in promoting the proliferation and effector functions of Th17 cells, which are characterized by the expression of the IL-17 family of cytokines [3]. Different psoriasis clinical manifestations, such as difficult-to-treat areas (scalp, palm, sole, flexural areas and nails), could sometimes also lead to modest clinical remission by using the most effective treatment [4]. In fact, there is still a need to clarify the different pathogenetic targets involved in the development of skin inflammatory diseases. Recently, Widiemann et al. demonstrated that immune genes promote differential susceptibility to inflammatory diseases, such as psoriasis and atopic dermatitis. Using single-cell RNA sequencing of human palm, sole and hip skin, it was possible to describe the distinguishing characteristics of palmoplantar and non-palmoplantar skin, revealing an altered immune environment in palmoplantar skin with downregulated and diverse immunological processes and decreased immune cell populations [9].

Tildrakizumab blocks immunological dysregulation with an indirect immunosuppression mechanism, which guarantees a significant and lasting improvement in skin psoriasis and, consequently, in the patients’ quality of life [10,17].

Based on the results of three years of treatment in two randomized and controlled phase-III clinical trials, reSURFACE 1 (NCT01722331) and reSURFACE 2 (NCT01729754), our real-life experience confirms the treatment’s safety and efficacy in maintaining clinical remission [14,15,16]. Several studies also analyzed different clinical practice settings of tildrakizumab-treated psoriasis patients [18,19,20,21,22,23,24,25,26,27].

Regarding the real-life setting evaluated by Burlando et al., the results revealed that tildrakizumab improves skin manifestations after only four weeks of treatment, suggesting a shorter clinical response with respect to those reported in registered clinical trials (RCTs). Moreover, tildrakizumab was as effective in overweight as in non-overweight patients, with similar responses in the naïve and non-naïve groups [8,17]. In addition, anti-IL-23 drugs such as guselkumab, risankizumab and tildrakizumab represent valid therapeutic options for multifailure psoriatic patients [19,20]. Furthermore, on difficult-to-treat sites, including the genitals, scalp, nail and palmoplantar areas, the drug had a favorable course in line with the data present in trials and real-life studies [21,22,23,24,25,26].

Psoriasis can considerably impact patients’ self-image, leading to embarrassment due to visible lesions, thereby resulting in low self-confidence, anxiety and depression. Patients receiving biological systemic treatments were observed to have more positive impacts to their quality of life compared to traditional systemic treatments [15,19].

In our real-life study, 53 patients with moderate-to-severe plaque-type psoriasis were enrolled. A prevalence of male patients (64.1%) and a mean psoriasis disease duration of 16 years were recorded. Descriptive analyses also showed that 37.7% of patients were affected by scalp psoriasis, and almost all of them had received different previous treatments.

The percentages of difficult-to-treat sites were as follows: 15.1% of patients had nail manifestations, 26.1% had palmoplantar psoriasis and a further 21.1% had psoriasis in the genital area. Considering the analyzed scores, our results show better responses in terms of PASI among patients younger than 35 y/o. This result could be linked to a reduced number of previous anti-psoriatic systemic treatments among young patients. On the other hand, we can suggest that an earlier administration of tildrakizumab could lead to a better prognosis and disease resolution. Several ongoing studies are focusing on the role of skin resident memory Th 17 cells and their need of local IL-23 to proliferate [20,28]. Our results can be considered real-life proof of this linkage, as tildrakizumab, by blocking IL-23p19, showed not only faster progress in young patients with less-established histories of disease

Regarding tildrakizumab’s efficacy on difficult-to treat-areas, we focused our analysis on nails and the palmoplantar area. In both sites, we observed a significant reduction in terms of NAPSI and PPPGA. However, whereas NAPSI decreased linearly through the weeks, PPPGA decreased dramatically mainly at W16. Our real-life study demonstrated that tildrakizumab is an effective option for the management of psoriasis in difficult-to-treat areas, as observed by Galluzzo et al. [22].

As already reported by Burlando et al. [8], our data shows that tildrakizumab promotes clinical remission in both naïve and non-naïve patients. We observed a slightly higher efficacy profile in naïve patients without statistical significance, but we believe that future real-life studies with larger samples could solve this issue. In our cohort, disease remission began at the end of the fourth week of treatment and remained constant from week 16 to week 52, which is in line with previous real-life studies [8]. Therefore, our real-life observations confirmed the registered clinical trials’ (RTI) reported data, showing the long-term efficacy of tildrakizumab use in psoriasis patients.

We also documented improvements in dysmetabolic conditions. Indeed, after 4 weeks of treatment, we observed a reduction of the mean fasting serum glucose and triglyceride levels, and their values continued to decrease throughout the entire period of enrollment. The effect of tildrakizumab on cardiometabolic parameters was already evaluated in a post hoc analysis of the two phase III trials reSURFACE 1 and reSURFACE2; however, even though decreases in fasting glucose and triglycerides were observed, the authors concluded that the changes in cardiometabolic disease risk factors following tildrakizumab treatment were limited [29]. Our data also reports a decrease in DLQI scores during tildrakizumab administration, confirming its role in ameliorating psoriatic patients’ quality of life. After 16–28 weeks of tildrakizumab treatment, DLQI (mean value) drastically decreased, and patients showed emotional and social improvements. Augustin et al. demonstrated in a phase-IV study of patient-reported well-being that tildrakizumab treatment improved the well-being of patients, their families and their healthcare professionals [30].

Interestingly, we found that patients with a DLQI score >10 also showed a BMI > 27, and 50% of them were affected by diabetes. These results confirm how comorbidities can have a great negative impact on the global burden of skin disease and in the quality of life of affected patients when considering their other diseases.

The presence of active comorbidities represents a crucial choice in psoriasis therapy at baseline because it could influence the patients’ final outcomes [31]. Studies comparing the efficacy and safety of biological therapies in patients with comorbidities are essential for informed shared decision-making in the treatment of psoriasis because the relationships between psoriasis treatment and comorbidities are still not elucidated [32]. Gottlieb et al. reported in their study that the most frequent comorbidities in moderate psoriasis include obesity, hypertension, PsA, hyperlipidemia and diabetes. Their post hoc analysis demonstrates that comorbidity status does not significantly affect the clinical responses of patients treated with secukinumab, except for patients with body weights of >90 kg [31].

In our patient cohort, the most prevalent comorbidity was hypertension (32.5%), followed by PsA (18.60%) and diabetes (13.9%). We stratified the patient population into two groups, those with and without comorbidities, evaluating the percentage decreases in PASI at different weeks of treatment. In these two groups of patients, we observed lower mean PASI reduction after 4 weeks of treatment in comorbid psoriasis patients, but during the remaining treatment period, the mean PASI decrease was similar in both groups.

Our results confirm the safety of tildrakizumab. In fact, we did not register any AEs that required discontinuation of treatment; the few AEs and the modality of administration increased patient adherence and satisfaction. Its versatility in the treatment of patients with moderate to severe burdens of disease and with the concurrent presence of comorbidities allow the consideration of tildrakizumab as a useful drug for fragile psoriatic patients. The presence of active comorbidities does not influence the safety and tolerability of tildrakizumab. In fact, many psoriasis patients suffer from diabetes, cardiovascular disease and infectious diseases and are not eligible for treatment with anti-TNF-α or anti-IL-17 drugs. As reported in the literature, anti-TNF therapies significantly increase the risk of infectious AEs, mainly the risk of severe infections and tuberculosis. It is known that anti-TNF therapies may affect hepatitis B, causing the reactivation and/or acceleration of preexisting liver damage. Consequently, screening for HBV and HCV markers (including HBsAg, anti-HBs, anti-HBc and anti-HCV) and the TB GOLD QuantiFERON test are mandatory in all patients who need anti-TNF therapy [32,33,34,35,36,37]. Furthermore, IL-17 inhibitors are known to increase patients’ risk of developing mycotic infections, such as *Pneumocystis jirovecii* pneumonia, histoplasmosis and candidiasis. The use of tildrakizumab should also be considered for patients with concomitant latent or treated TBC [17,38]. Tildrakizumab continued to be generally well-tolerated over the longer-term in the reSURFACE extension studies [14,39,40,41].

## 4. Materials and Methods

### 4.1. Enrolled Patients and Study Design

We evaluated fifty-three psoriatic patients receiving subcutaneous injections of tildrakizumab at a dosage of 100 or 200 mg (for patients weighing > 90 kg) at weeks 0 and 4 followed by 100 mg every 12 weeks. Patients over 18 years old with moderate-to-severe psoriasis and who were eligible for systemic treatment were included in the study. Patients who presented the involvement of difficult-to-treat areas such as the face, scalp, hands or genital areas and whose conditions failed to improve with treatment with DMARDs, biological therapies and small molecules were also considered eligible for treatment with tildrakizumab. The following data were available at baseline: age, sex, comorbidities and previous treatments. All patients signed informed consent for their participation in the study and for publication. The Declaration of Helsinki was respected throughout the study.

### 4.2. Score and Laboratory Analyses Evaluation

The assessments used include the Psoriasis Area and Severity Index (PASI), Dermatology Life Quality Index (DLQI) and, if applicable, the Nail Psoriasis Severity Index (NAPSI) and Palmoplantar Psoriasis Physician Global Assessment (PPPGA). PASI, DLQI, NAPSI and PPPGA scores were recorded at each visit. Laboratory tests (complete blood count, erythrocyte sedimentation rate, C-reactive protein, alanine aminotransferase, aspartate aminotransferase, gamma-glutamyltransferase, creatinine, urea nitrogen, high-density lipoprotein and low-density lipoprotein cholesterol, triglycerides, glucose, uric acid, lactate dehydrogenase and creatine phosphokinase) were performed at baseline and at each visit, whereas QuantiFERON-TB gold tests and serology for hepatitis B virus, hepatitis C virus and human immunodeficiency virus were performed at baseline and at week 48.

### 4.3. Safety

The safety and tolerability of tildrakizumab were evaluated over the duration of the study (including the presence of any adverse events (AEs)). Safety and tolerability were investigated by examining AEs as reported by patients during follow-up examinations at certain weeks of treatment, as reported by Galluzzo et al. [22].

### 4.4. Statistical Analysis

Data were collected in a single database and were first used for the descriptive statistics of all parameters of interest, then used for inferential statistics. The results are expressed as means or percentages, considering the type of each analyzed variable. In order to analyze the differences in terms of PASI and DLQI before and after treatment at different timepoints (weeks), the *t*-test or ANOVA was used, followed by the Bonferroni correction ad hoc test. The non-parametric Spearman rank test was used for correlation comparations, expressed by the rho coefficient (0–1). Results were considered statistically significant for *p*-values < 0.05 [42]. The software used for statistical analysis was SPSS v.20 (Chicago, IL, USA).

## 5. Limitations

Our study has several limitations. This is a real-life study; thus, it is missing data that would typically be found in a retrospective/observational study. At week 52, not all patients were examined (*n* = 34). In our cohort, 47.2% of patients were biologically naïve, and this may have improved the observed response to tildrakizumab. For biochemical analyses on patients with cardiometabolic comorbidities, we analyzed only 30 patients (30/36), and we only analyzed 7 patients without comorbidities (7/10). A standardized protocol should be drawn up.

## 6. Conclusions

In conclusion, in our real-life study, the use of tildrakizumab favored crucial clinical improvements in treated patients. In addition to evaluating the scores normally used in assessing the clinical improvement of patients, we also focused on the comorbidities and psychological implications of enrolled patients to highlight a link between them and skin diseases with metabolic/vascular involvement. In fact, most of the patients enrolled in the study had a BMI compatible with pre-obesity, beyond diabetes or hypertension, and a DLQI > 10. Further long-term studies are underway to evaluate all of the specific characteristics of the drug so that it can be used with increasing safety on a large number of patients. Furthermore, administration every 12 weeks and its efficacy even in difficult-to-treat sites makes tildrakizumab a valid therapeutic option, mainly in patients whose conditions failed to improve with other biologics or traditional systemic therapies and in patients with comorbidities. Future studies should be carried out both to detect the correlation between different diseases such as psoriasis and diabetes and/or cardiovascular diseases and to add new insights regarding the mechanism of action of tildrakizumab beyond its effect on psoriasis.

## Figures and Tables

**Figure 1 pharmaceuticals-16-00526-f001:**
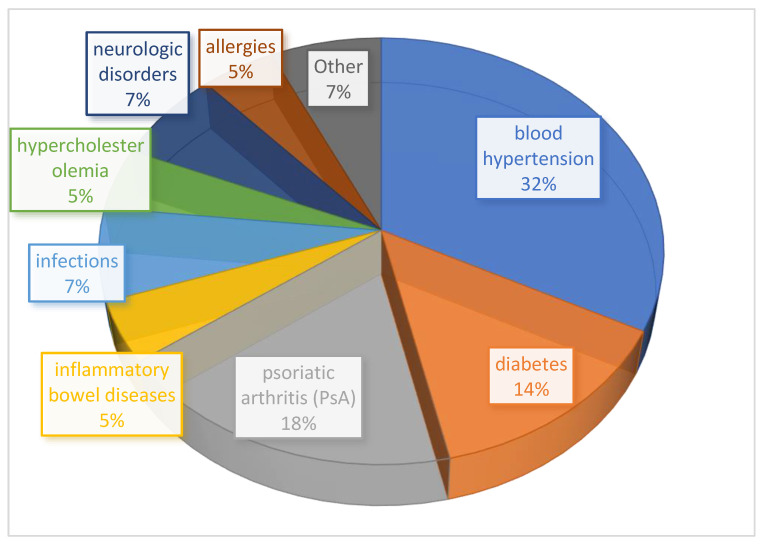
Distribution of comorbidities in enrolled patients.

**Figure 2 pharmaceuticals-16-00526-f002:**
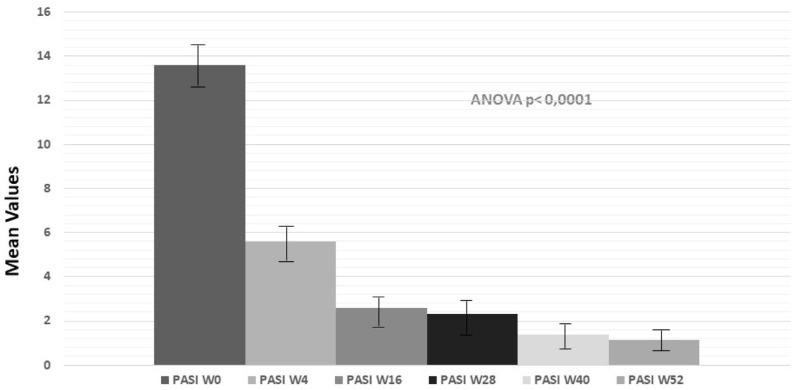
Mean PASI (mPASI) reduction at weeks 4, 16, 28 and 52 (Anova test; *p* < 0.0001).

**Figure 3 pharmaceuticals-16-00526-f003:**
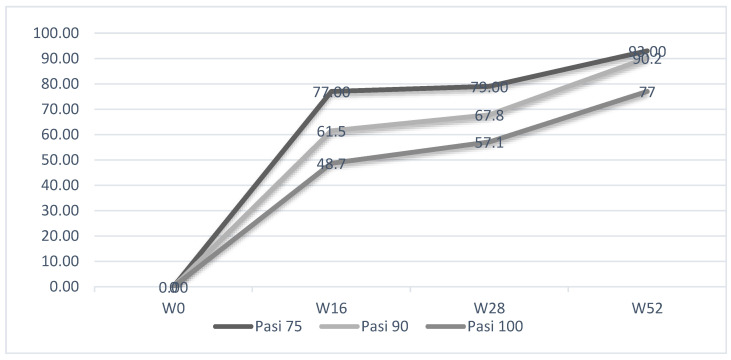
Mean PASI (mPASI) reduction and percentages of patients achieving PASI 100/90/75 at weeks 4, 16, 28, and 52.

**Figure 4 pharmaceuticals-16-00526-f004:**
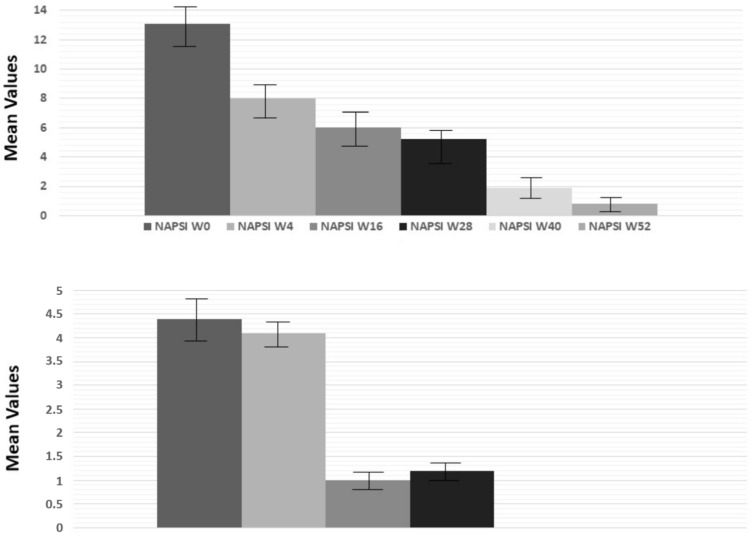
Evaluation of NAPSI and PPPGA indices (ANOVA test; *p* < 0.001 and *p* < 0.0001, respectively up and down) at weeks 4, 16, 28 and 52 compared to baseline.

**Figure 5 pharmaceuticals-16-00526-f005:**
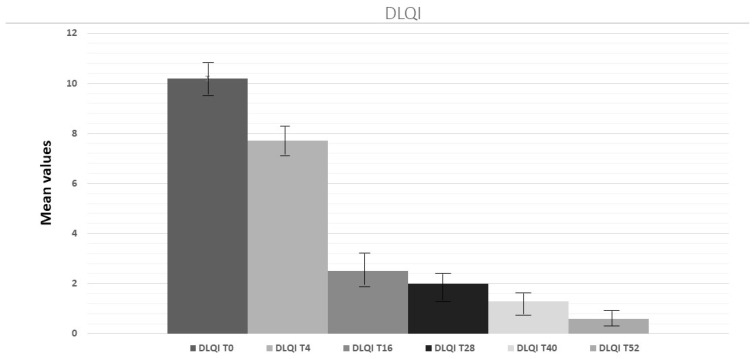
Evaluation of DLQI index at weeks 4, 16, 28 and 52 compared to baseline.

**Figure 6 pharmaceuticals-16-00526-f006:**
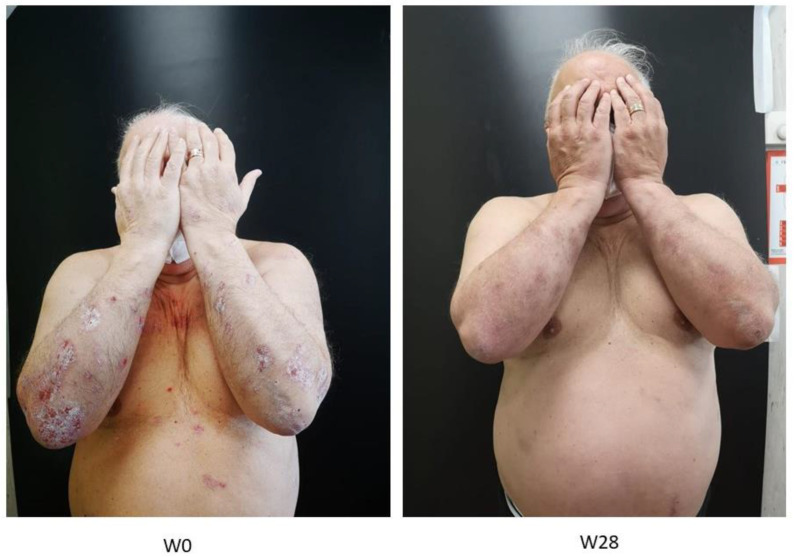
Clinical evaluation at W0 and after 28 weeks of treatment.

**Figure 7 pharmaceuticals-16-00526-f007:**
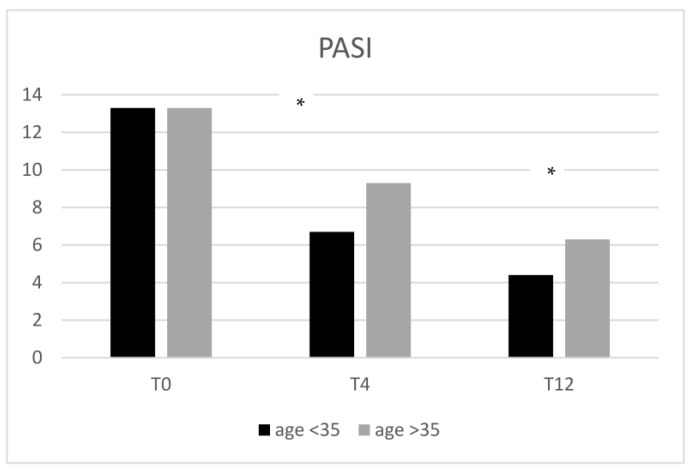
PASI comparison between patients with ages of onset <35 and >35 years of age. Patients with an age of onset < 35 showed better PASI reduction compared to patients with an age of onset > 35 years (ANOVA, *p* < 0.001; * *t*-test *p* < 0.01).

**Figure 8 pharmaceuticals-16-00526-f008:**
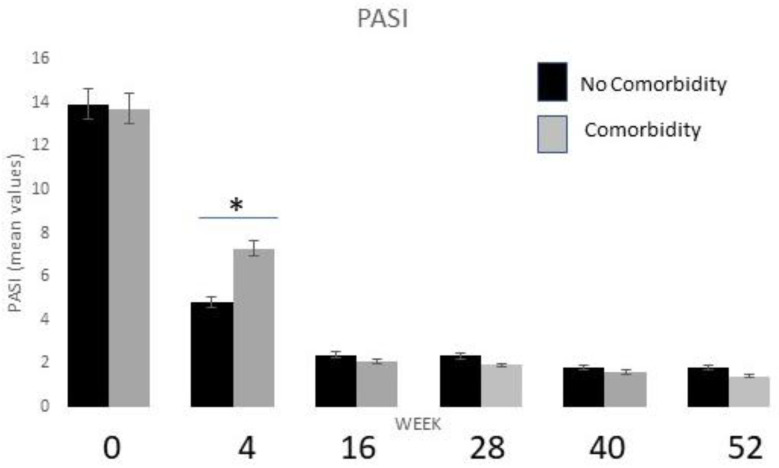
Evaluation of PASI mean values for patients with or without active comorbidities at baseline and at different timepoints of tildrakizumab treatment (* *p* < 0.01).

**Table 1 pharmaceuticals-16-00526-t001:** Clinical and demographic features of patients (patient number = 53).

	*n* (%) 53 (100%)
Mean age, yrs: 51.2 (range 22–79)	
Male	34 (64.1%)
Female	19 (35.9%)
Smokers	25 (47.1%)
Family history	24 (45.2%)
Disease duration	16 yrs (range 1–46)
Mean age of onset, yrs: 34.6 (range 8–73)	

**Table 2 pharmaceuticals-16-00526-t002:** Previous treatments of enrolled patients.

Previous Treatments	*n* (%) 77 (100)
Topical treatments	10 (12.9)
Methotrexate	14 (18.2)
Cyclosporine	21 (27.3)
Acitretin	3 (3.9)
NB-UVB	4 (5.2)
Dimethyl fumarate	3 (3.9)
Apremilast	8 (10.4)
Adalimumab	2 (2.6)
Ustekinumab	2 (2.6)
Secukinumab	7 (9.1)
Others (etanercept, brodalumab)	3 (3.9)

## Data Availability

All data generated or analyzed during this study are included in this article. Further enquiries can be directed to the corresponding author.

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
