# Peer review of "A Real-Life Study on the Use of Tildrakizumab in Psoriatic Patients"

_pharmaceuticals, 2023, doi:10.3390/ph16040526_

Round 1

Reviewer 1 Report

The authors reported the results of a real-life clinical experience of tildrakizumab in patients affected by psoriasis and comorbidities. The manuscript is interesting and well-written. However, I have some suggestions.
My comments:

- Abstract: before use abbreviations (e.g.PASI) you should report the words you are abbreviating

- Abstract: the dosage of tildrakizumab should include the induction phase

- Introduction: before use abbreviations (e.g.IL) you should report the words you are abbreviating

- Introduction: introduction section is too long. Please syntetize

- Introduction: the introduction of biologics revolutionized the management of psoriasis (10.1080/14712598.2022.2089020)

- Material and Methods: this section should be reported before the results

- Results: I really appreciated this section. Please include more data about the safety.

- Discussion: tildrakizumab showed to be a useful weapon also in severe forms of psoriasis (e.g. 10.1111/dth.15030)

- Discussion: Other real-life experiences should be briefly discussed (10.1111/jdv.18594 - 10.4081/dr.2022.9447 - 10.1111/dth.15941)

Author Response

See attached , please, thank you.

Reviewer 2 Report

- Could please specify what are "serious adverse events" on the introduction section
- How are demografic data distributed? At least interquartile ranges should be laid out
- Please consider the following 3 articles in your discussion:

doi: 10.1111/jdv.18594.

doi: 10.1136/bmjopen-2021-060536.

doi: 10.1111/jdv.18572

-About stratification and data analysis and discussion: how do you believe that known prognostic factors and demografic factors would influence both clinical and biochemical response to Tildrakizumab in a multivaried analysis (i.e. affection difficult to treat areas or being bio-experienced as displayed in limitation section)
- In order to have a clearer portray of the clinical relevance of your data, with regards to "real life", could please go into depth of how your intention to treat analysis would compare to a per protocol analysis

Author Response

See attached, please, thank you.

Reviewer 3 Report

The paper presents real world data on the use of tildrakizumab in the treatment of psoriasis.  However, a few points need improvement:

  • the abstract lacks any quantitative data;
  • the aim of the study is not clearly stated
  • Table 1 - please provide range for age of onset and disease duration (not only mean value)
  • what percentage of patients were overweight and what were obese?
  • specify "others" in table 2
  • a mean glucose value above 200 mg/dl at entry into the study would indicate that most patients are diabetic. Is this the right data, was the test taken after fasting?  if the patient had glucose above 200, did he receive hypoglycemic treatment?
  • Title suggest analysis of tildrakizumab wfficacy in patients with psoriasis and concomitant baseline comorbid conditions. However there are no data in the manuscript. 
  • Please add more current study on real world data for tlidrakizumab in psoriasis

Author Response

See attached, please, thank you.

Round 2

Reviewer 1 Report

All the changes have been made. The authors improved the quality of the manuscript. Now, the article is suitable for the publication.

Author Response

Thank you for your positive opinion.

Reviewer 3 Report

The authors took into account most of the reviewer's comments.

Two issues remain:

  • biochemical parameters - an average glucose value above 200 mg/dl is a very high value suggesting diabetes. In response, the authors wrote that they do not know whether the blood was collected after fasting, or whether the patients were additionally on a diet or hypoglycaemic drugs. The data are misleading because they suggest normalization of glycemia during therapy - with an effect that even the best registered antidiabetic  drugs do not have. I suggest removing this data, as it affects the credibility of the entire work.
  • the title suggests that the paper will include data on tildrakizumab in patients with comorbidities, but there is no data comparing the group of psoriasis without comorbidities, with one and several diseases. Without this the work is no different from other real world data for tildrakizumab and therefore „comorbidities” should be removed from the title.

Author Response

Dear reviewer, thank you for your suggestions.

Regarding the graph on glucose and triglycerides (figure 8), we would like to underline, as reported in the text (gray sentence), that includes all patients with all comorbidities, and their overweight status .

 We reported number of patients analysed. (n=30/36 with cardiometabolic comorbidities; number total comorbidities patient=43).  In the whole group of patients enrolled, seven of them without comorbidities were also evaluated for the biochemical parameters without detecting anomalies (no significant differences were recorded).

“Patients’ comorbidities were also recorded with blood hypertension as the most frequent (32.5%), followed by diabetes (13.9%), infections (7%), neurologic disorders (7%), hypercholesterolemia (4.65%), inflammatory bowel diseases (4.65%), allergies (4.65%), while 7% were affected by other pathologies (Figure 1).”

“Patients showed a Body Mass index (BMI) mean value >27.2 (pre obesity status), whereas 12 of 53 enrolled patients were overweight (BMI 26-29.99) and 16 were obese with a BMI>30 (range 30-39)”

In addition, other studies have already addressed the effect that the drug could have on other diseases, such as diabetes or cardiovascular diseases, as reported in the manuscript in several points.

“The effect of tildrakizumab on cardiometabolic parameters was already evaluated in a post hoc analyses of the two phase 3 trials ReSURFACE 1 and ReSURFACE2, however, even though decreases in fasting glucose and triglycerides were observed,”

  1. Reich K., Warren R. B., Iversen L., Puig L., Pau-Charles I., Igarashi A., et al. Long-term efficacy and safety of tildrakizumab for moderate-to-severe psoriasis: pooled analyses of two randomized phase III clinical trials (reSURFACE 1 and reSURFACE 2) through 148 weeks. Br J Dermatol. 2020 Mar;182(3):605-617. doi: 10.1111/bjd.18232. Epub 2019 Jul 18.
  2. Ricceri F, Chiricozzi A, Peris K, Prignano F. Successful use of anti-IL-23 molecules in overweight-to-obese psoriatic patients: A multicentric retrospective study. Dermatol Ther. 2022 Nov;35(11):e15793. doi: 10.1111/dth.15793. Epub 2022 Sep 9. PMID: 36038527.
  3. Menter MA, Mehta NN, Lebwohl MG, Gottlieb AB, Mendelsohn AM, Rozzo SJ, Leonardi C. The Effect of Tildrakizumab on Cardiometabolic Risk Factors in Psoriasis by Metabolic Syndrome Status: Post Hoc Analysis of Two Phase 3 Trials (ReSURFACE 1 and ReSURFACE 2). J Drugs Dermatol. 2020 Aug 1;19(8):703-708. doi: 10.36849/JDD.2020.5337. PMID: 32845115.
  4. Narcisi A, Valenti M, Gargiulo L, Ibba L, Amoruso F, Argenziano G, Bardazzi F, Burlando M, Carrera CG, Damiani G, Dapavo P, Dini V, Franchi C, Girolomoni G, Guarneri C, Loconsole F, Sampogna F, Travaglini M, Malagoli P, Costanzo A. Real-life effectiveness of tildrakizumab in chronic plaque psoriasis: A 52-week multicentre retrospective study-IL PSO (Italian landscape psoriasis). J Eur Acad Dermatol Venereol. 2023 Jan;37(1):93-103. doi: 10.1111/jdv.18594. Epub 2022 Oct 5. PMID: 36156312.

We believe that the focus on comorbidities in our cohort psoriasis patients is crucial because already in several studies a beneficial role of the drug has been reported on other diseases that share a common origin with psoriasis. Our study further adds useful and more detailed real-life data in terms of long-term efficacy and safety in these complex patients also affected by comorbidities that increase the burden of psoriasis and make remission more difficult.

So, we prefer not to remove it from the title.

However, considering your advice, we include this concern among the limits. Certainly, these deepening could be observed with greater attention also taking into account all the pharmacological treatments of patients with comorbidities.

Round 3

Reviewer 3 Report

The presented comorbidities occur with a similar frequency as in other populations of psoriatic patients from clinical trials or real world data. If the authors insist that their work is focused on showing the impact of comorbidity on drug response, then this should be reflected in the results. In studies that have such a goal, the response and safety profile of the drug are most often compared depending on the number of comorbidities.

As an example I recommend article: 

Clinical efficacy and safety of secukinumab in patients with psoriasis and comorbidities: pooled analysis of 4 phase 3 clinical trials.

Gottlieb AB, et al. J Dermatolog Treat. 2022.PMID: 33023357 Clinical Trial.

In addition, the presentation of data on the effect of the drug on glycemic values, without information about the time of collection (after fasting?), possible diagnosis of diabetes, change in diet, treatment - it cannot be argued that this is only the effect of a biological drug. This is deliberately misleading. The reduction of the average glycemic value from 200 mg/dl to 100 mg/dl is not seen with the best antidiabetic drugs. If this was due only to the effect of the drug, there would be a risk of hypoglycaemia, and such side effects have not been reported in this cohort.

Author Response

“The presented comorbidities occur with a similar frequency as in other populations of psoriatic patients from clinical trials or real world data. If the authors insist that their work is focused on showing the impact of comorbidity on drug response, then this should be reflected in the results. In studies that have such a goal, the response and safety profile of the drug are most often compared depending on the number of comorbidities”.

As an example I recommend article: Gottlieb AB, Wu JJ, Griffiths CEM, Marfo K, Muscianisi E, Meng X, Frueh J, Lebwohl M. Clinical efficacy and safety of secukinumab in patients with psoriasis and comorbidities: pooled analysis of 4 phase 3 clinical trials. J Dermatolog Treat. 2022 May;33(3):1482-1490. doi: 10.1080/09546634.2020.1832187. Epub 2020 Oct 21. PMID: 33023357.

Dear Reviewer,

Thank you for your comments.

We accepted your suggestions and we modified the manuscript in results section with a new bar graph that shows the stratification of patients in two groups, with and without active comorbidities. The data were completed with the evaluation of mean PASI decrease during the entire period of treatment with tildrakizumab. We have also included the data reported in the paper by Prof. Gottlieb et al in the discussion.

“In addition, the presentation of data on the effect of the drug on glycemic values, without information about the time of collection (after fasting?), possible diagnosis of diabetes, change in diet, treatment - it cannot be argued that this is only the effect of a biological drug. This is deliberately misleading. The reduction of the average glycemic value from 200 mg/dl to 100 mg/dl is not seen with the best antidiabetic drugs. If this was due only to the effect of the drug, there would be a risk of hypoglycaemia, and such side effects have not been reported in this cohort.”

Dear Reviewer, thank you for your comments.

On the basis of your suggestions, we have decided to remove from the manuscript the observations regarding the glycaemic parameters of our patients. Although we have collected these parameters that seemed interesting throughout the period of drug treatment, also in relation to the improvement of psoriasis, as you rightly observed, it would be necessary to list other anamnestic data on lifestyle, eating habits and antidiabetic drugs.

We thank you for your suggestions and hope that the edited manuscript is now improved.

 Elena Campione and co-authors

 Prof. Elena Campione

Dermatology Unit

Department of Systems Medicine

University of Rome Tor Vergata

Via Montpellier 1

Rome, Italy